# Genetic Etiology of Nonsyndromic Hearing Loss in Hungarian Patients

**DOI:** 10.3390/ijms24087401

**Published:** 2023-04-17

**Authors:** Margit Pál, Dóra Nagy, Alexandra Neller, Katalin Farkas, Dóra Leprán-Török, Nikoletta Nagy, Dalma Füstös, Roland Nagy, Adrienne Németh, Judit Szilvássy, László Rovó, József Géza Kiss, Márta Széll

**Affiliations:** 1Department of Medical Genetics, Albert Szent-Györgyi Medical School, University of Szeged, 6725 Szeged, Hungary; 2ELKH-SZTE Functional Clinical Genetics Research Group, Eötvös Loránd Research Network, 6720 Szeged, Hungary; 3Institute of Medical Genetics, Kepler University Hospital Med Campus IV, Johannes Kepler University Linz, 4020 Linz, Austria; 4Department of Oto-Rhino-Laryngology and Head-Neck Surgery, Albert Szent-Györgyi Medical School, University of Szeged, 6720 Szeged, Hungary; 5Department of Oto-Rhino-Laryngology and Head-Neck Surgery, University of Pécs, 7621 Pécs, Hungary; 6Department of Otorhinolaryngology and Head-Neck Surgery, University of Debrecen, 4032 Debrecen, Hungary

**Keywords:** genetic hearing loss, cochlear implants, genetic testing

## Abstract

Hearing loss is the most prevalent sensory disorder worldwide. The majority of congenital nonsyndromic hearing loss (NSHL) cases are caused by hereditary factors. Previously, the majority of NSHL studies focused on the *GJB2* gene; however, with the availability of next-generation sequencing (NGS) methods, the number of novel variants associated with NSHL has increased. The purpose of this study was to design effective genetic screening for a Hungarian population based on a pilot study with 139 NSHL patients. A stepwise, comprehensive genetic approach was developed, including bidirectional capillary sequencing, multiplex ligation-dependent probe amplification (MLPA), and an NGS panel of 108 hearing loss genes. With our results, a genetic diagnosis was possible for 92 patients. Sanger sequencing and MLPA identified the genetic background of 50% of these diagnosed cases, and the NGS panel identified another 16%. The vast majority (92%) of the diagnosed cases showed autosomal recessive inheritance and 76% were attributed to *GJB2*. The implementation of this stepwise analysis markedly increased our diagnostic yield and proved to be cost-effective as well.

## 1. Introduction

Hearing loss (HL) is the most prevalent sensory disorder, affecting millions of people worldwide with an incidence of 0.1–0.2% in the newborn population. HL in children is classified as prelingual or postlingual, depending on the age of onset. Most congenital HL cases are caused by hereditary factors; whereas, most HL that develops in school-aged children and adolescents is acquired. Hereditary HL appears as nonsyndromic hearing loss (NSHL); which, is associated with more than 100 genes, or manifests as part of a syndrome (syndromic hearing loss, SHL). There are more than 400 genetic syndromes that present with HL [1]. Some HL genes are associated with both NSHL and SHL. The genetic overlap between NSHL and SHL can make differential diagnosis complicated. 

Prelingual NSHL disorders are inherited as autosomal recessive in 80% of the cases and as autosomal dominant in 15–20%. In the remaining cases, the X-linked or mitochondrial inheritance is observed. For postlingual NSHL, most cases are reported as autosomal dominant [2]. 

Considering the complexity of hearing, it is not surprising that many genes are involved in the process. In fact, it is estimated that 1% of all human genes are associated with hearing [3]. Previously, most genetics studies of NSHL focused on the connexin 26 protein (encoded by the *GJB2* gene), as it is a key component of the gap junction in the cochlea. Next-generation sequencing (NGS) methods have improved the identification of causative variants for HL, as well as for many other hereditary disorders. To date, 124 genes have been associated with monogenic NSHL (https://hereditaryhearingloss.org/, accessed on 22 January 2023).

The prevalence of gene variants associated with NSHL varies among different ethnic groups. In patients of Caucasian origin, only a few of the 124 identified genes account for the majority of NSHL cases. In autosomal recessive cases, *GJB2* and *STRC* are the most frequent causative genes, accounting for approximately 50% and 16% of the cases, respectively. Only a few genes (*OTOF, MYO7A, MYO15A*) have a contribution higher than 2%. In autosomal dominant cases, the picture is more diverse. The most common variants are from the *MYO6* gene, accounting for 22%, *TECTA* for 18%, *ACTG1* and *WFS1* each for 9%, *POU4F3* for 6.5%, *MYO7A* for 5%, *MYH14* and *COL11A2* each for 4% [4]. These data were collected in 2021 by Castillo and colleagues from several studies performed in different countries at different times. The frequency of the reported genes might be influenced by ethnic factors and is limited to the number of causative genes known at the time of publication.

The purpose of this study was to screen Hungarian patients with NSHL and to develop the most effective screening scheme for this disease. For this, we used a comprehensive, stepwise, sequencing-based genetic testing approach. 

## 2. Results

### Genetic Screening

Using our stepwise testing approach (Figure 1), genetic diagnosis was possible for 92 patients, corresponding to a total diagnostic yield of 66% (92/139). The diagnostic yield for Sanger sequencing and MLPA was 50% (70/139) and the NGS panel identified another 16% (22/139). In the prelingual NSHL group, the diagnostic yield was 73% (76/105) and, for the postlingual NSHL group, 47% (16/34). In 26 cases (19%), no variant was identified (13% (14/105) in the prelingual group and 35% (12/34) in the postlingual group, Figure 2). This difference was statistically significant (*p* = 0.006674), indicating that it is more likely to establish the genetic diagnosis in prelingual NSHL than in postlingual NSHL patients. 

The vast majority of the diagnosed cases showed autosomal recessive inheritance, 92% (85/92). A further 7% (6/92) of the cases were autosomal dominant, and 1% (1/92) were X-linked (Figure 3A). The distribution of *GJB2* and non-*GJB2* cases was 76% (70/92) and 24% (22/92), respectively (Figure 3B). 

For the 92 diagnosed patients, a total of 200 causative/probably causative variants were identified for 22 different genes (Figure 3C). These 200 variants included 54 unique variants, of which 33 were already known recurrent pathogenic variants and 21 were novel. Of the 200 variants, 142 (76%) were frameshift, 36 (19%) missense, 12 (6%) nonsense, 8 (4%) splice site, and 2 (1%) copy number variants (CNV) (Table 1 and Table 2; Appendix A).

A total of 181 variants were classified as pathogenic or likely pathogenic and 19 as variants of unknown significance (VUS) or VUS with a tendency toward likely pathogenic (VUS/LP) based on the homozygosity, co-occurrence of a second variant in the same recessive gene, genotype–phenotype comparison, or segregation analysis (Table 1 and Table 2; Appendix A).

For 21 (15%) patients, genetic diagnosis was inconclusive (Figure 2), as only one variant was identified in autosomal recessive genes or one variant was identified in genes with both autosomal dominant and recessive inheritance, where the genotype–phenotype comparison could not make a clear diagnosis. For seven cases, the variant occurred in the *GJB2* gene. In the rest of the cases, the variant occurred in one of the following genes: *WFS1, MYO7A, CDH23, OTOF, PNPT1, PTPRQ, USH1C, STRC.* One case had variants for both the *GJB2* and *OTOA* genes. For these patients, an additional causative variant could not be detected, even with the extended NGS analysis. 

In two cases, variants were identified in more than one HL gene. In the first case, an *OTOA* whole gene duplication was identified in addition to a heterozygous *GJB2* variant, mentioned above. In the second case, a *TJP2* heterozygous variant was detected in addition to a homozygous *SLC26A4* variant (Table 1 and Table 2; Appendix A). 

Thirteen pathogenic variants were identified in the *GJB2* gene (Table 1). The most common variant was c.35delG; which, was identified in homozygous form for 57 patients and in compound heterozygous form with seven different other pathogenic variants (c.101T>C; c.139G>T; c.167delT; c.313_326del; c.439G>A; c.551G>C; c.−23+1G>A) for 11 patients. All patients carrying the c.35delG variant in the homozygous form had prelingual onset of NSHL and received a cochlear implant. Compound heterozygous variants, other than the c.35delG, were detected in three patients: c.[109G>T];[269T>C], c.[109G>T];[427C>T] and c.[427C>T];[c.−23+1G>A]. For eight patients, only one pathogenic *GJB2* variant was identified: c.35delG in four cases; c.71G>A in one case; c.101T>C in two cases; and c.119C>A in one case (Table 1; Appendix A). No variants were identified in the *GJB3* and *GJB6* genes. 

Non-*GJB2* variants identified with the NGS panel and MLPA are listed in Table 2. Causative or probable causative variants with autosomal dominant inheritance were identified in 6 cases: *KCNQ4* in two cases and *WFS1*, *TECTA, POU4F3,* and *OSBPL2* in one case each (Table 2, Figure 3C). 

Causative or probable causative variants with autosomal recessive inheritance were identified in 15 cases. The most frequently identified gene was *MYO15A,* with five pathogenic/LP and one VUS variant from three patients. Three probable causative variants were identified in the *TMC1* and *SLC26A4* genes, each from two patients. Two potentially causative mutations were identified in *CDH23, TMPRSS3* from two patients, and in *COL11A2* from one patient. One pathogenic variant was identified in the *MARVELD2* and *PTPRQ* genes from two patients, and, in both cases, the variant was homozygous. A hemizygous variant was detected in the X-linked recessive *POU3F4* gene from one male patient (Table 2 and Figure 3, Appendix A).

## 3. Discussion

Previous genetic analyses of patients with NSHL in Hungary mainly focused on examining the *GJB2* gene [7,8], as this gene is the most common NSHL-causing gene worldwide. However, the distribution of *GJB2* variants varies between populations. In Caucasians, the most frequent variant is the c.35delG; which, may account for up to 70% of alleles in some populations [9]. In our cohort of 139 Hungarian NSHL patients, *GJB2* pathogenic alterations were found to be causative in 70 cases (50%; which, is close to previous results with another Hungarian cohort (55%) [10] and lies within the European prevalence range (11.7–57.5%) [4]. In addition to the c.35delG variant, twelve other recurrent pathogenic variants were detected, and, of these, ten variants have already been identified in previous studies involving Hungarian NSHL patients [7,8] and two variants were detected for the first time in Hungary in our study (Table 1). Similar to other studies, the c.35delG homozygous and compound heterozygous cases, except for the c.[35delG];[101T>C] genotype, were associated with profound prelingual NSHL. The c.[35delG];[101T>C] genotype was associated with a somewhat milder NSHL. This observation is consistent with the results of a previous study [11]. The c.101T>C; p.Met34Thr variant is likely to be disease-causing only if it is in a homozygous state or co-occurs with another pathogenic mutation.

The *GJB*2 c.−23+1G>A splice site mutation was reported as a significant contributor to NSHL (up to 5%) in many European countries [4] and in the Hungarian cohorts (3.1%) [10]. However, we identified this variant only in three compound heterozygous cases (2.2%). The difference may be explained by our smaller sample size. Two patients carrying the c.[35delG];[−23+1G>A] genotype and the c.[427C>T];[−23+1G>A] genotype presented severe and profound prelingual NSHL. 

We found only one heterozygous variant in the *GJB2* gene in eight patients. We suppose that these detected variants alone are probably not responsible for the HL. This is likely an incidental finding, since the allele frequency is high in the general population for this gene: c.35delG alone accounts for approximately 0.9–1.2% in European populations [5,12]. Furthermore, heterozygous carriers have been reported that have impaired hearing at very high frequencies (8000–10,000 Hz) with aging, but that, unlike the patients in our study, never completely lacked a DPOEA response [13]. For one patient with the *GJB2* c.35delG variant, another recessive heterozygous variant of the *OTOA* gene was also identified. To our knowledge, the digenic inheritance of heterozygous variants in these two recessive genes has not yet been described and thus a causality cannot be deduced. 

No variants were identified for the *GJB6* and *GJB3* genes in our study, despite the relatively high frequency of 8.2% for variants in these genes in Western European countries [4]. The common del(GJB6-D13S1830) deletion truncates not only the *GJB6* gene itself but removes a large region close to the *GJB2* gene and eliminates a distinct *cis*-acting regulatory element that is essential for the normal expression of the *GJB2* gene in the inner ear [14]. *GJB6* deletions in combination with *GJB2* mutations are thought to be causative—either by the removal of the *GJB2* regulatory element or by digenic inheritance. However, our results are consistent with the previous studies from Hungary [10]: both *GJB6* deletions and *GJB3* alterations are rare causes of NSHL in Hungary (0.48%). 

With the addition of MLPA analyses, we were able to detect two non-*GJB2* variants, and, with our custom-designed NGS NSHL panel, a further 35 non-*GJB2* variants. These results enabled us to establish a rare genetic diagnosis in 22 further cases, increasing our diagnostic yield by 16% (Figure 1). All the identified *GJB2* variants are known variants; whereas, 21 non-*GJB2* variants identified in this study are, to our knowledge, novel. 

The distribution of the mode of inheritance in our NSHL cohort was similar to that previously described in the literature [2]: 92% autosomal recessive vs. 80% previously reported, 7% autosomal dominant vs. 20%, and 1% X-linked vs. 1% (Figure 3). 

Although the second most frequently identified NSHL gene in Europe is the stereocilin-coding *STRC* gene (OMIM #603720, DFNB16), which may make up to 16% of all cases [4], we were not able to identify any biallelic disease-causing variants for this gene. The majority (99%) of *STRC* causative variants are large CNVs [15]. In our NSHL cohort, only one heterozygous and contiguous gene deletion encompassing the *STRC* gene was detected. This discrepancy may be explained by the fact that *STRC* mutations are most common among patients with mild-to-moderate NSHL, and our cohort consisted mostly of patients with profound NSHL. 

The second most frequent gene identified in our cohort was *MYO15A*. Six different variants of *MYO15A* were identified in three patients (2%). *MYO15A* is the largest gene in the study, consisting of 66 exons and encoding a protein of 3,530 amino acids. The encoded protein plays a vital role in the elongation and development of stereocilia and actin filaments. *MYO15A* is the second most common gene in the European NSHL population (OMIM: #600316, DFNB3), affecting roughly 11% of NSHL patients [4]. All our patients carrying *MYO15A* variants suffered from congenital and profound NSHL.

Three different variants were identified in *SLC26A4* gene and also three different variants in *TMC1* gene. Two patients carried the *SLC26A4* variants (one homozygous and one compound heterozygous) and two patients the *TMC1* variants (also one homozygous and one compound heterozygous) (Table 2). *SLC26A4* and *TMC1* genes both comprised 1.4% and 1.4% of our patient cohort. Variants in these two genes were detected in Europe with a frequency of 3.8% and 5.1%, respectively [4]. The *SLC26A4* gene encodes a transmembrane protein known as pendrin; which, transports Cl^−^, I^−^, and HCO^3−^ anions. Most pendrin expression is found in the inner ear, thyroid gland, and kidney. *SLC26A4* mutations are either responsible for autosomal recessive NSHL with enlarged vestibular aqueduct (OMIM #600791) or Pendred syndrome (OMIM #274600) [16] accompanied by hypothyroid goiter and increased risk for thyroid cancer. Hypothyroidism was not reported in our patients carrying *SLC26A4* variants. The *SLC26A4* c.349C>T and c.1204G>T variants have been reported in the literature [17]; the third c.1670G>T variant, to our knowledge, is novel. HL started in the early postlingual period for both patients carrying *SLC26A4* variants. They both responded well to hearing aids at an early age, but, due to progression, a cochlear implantation was performed at 7 and 10 years of age. The *TMC1* gene plays an important role in the function of hair cells: this gene encodes a 6-pass integral membrane protein, which is a component of mechanotransduction channels in the hair cells of the inner ear [18]. Variants of the gene can cause autosomal recessive (OMIM #600974, DFNB7) or autosomal dominant deafness (OMIM #606705, DFNA36). The *TMC1* c.2030T>C variant was identified in homozygous form in one of our patients. This variant was previously reported in homozygous form in a Turkish family [19]. 

Although our sample size was moderate, one novel variant in the *PTPRQ* gene (c.5959C>T) was detected in three unrelated individuals (one heterozygous and two homozygous). This unusually high frequency of a novel variant could be due to an unknown relationship between the examined individuals, consanguinity, or founder effect for the Hungarian population. Furthermore, two recurrent variants in two genes were detected in overall four unrelated patients: the splice variant c.1331+2T>C in the *MARVELD2* gene and the frameshift variant c.208delC in the *TMPRSS3* gene. The *MARVELD2* c.1331+2T>C is a known variant in the Pakistani and Roma populations [20]. The *TMPRSS3* c.208delC variant was first reported in Spanish and Greek patients [21]. *TMPRSS3* mutations are predicted to account for <1% of NSHL in Caucasians [21]; but, are more frequent among Pakistani (1.8%) and very frequent in Turkish patients (23%) [16]. 

*WFS1* variants were identified in five patients. WFS1 participates in the regulation of cellular Ca^2+^ homeostasis by, at least partly, modulating the filling state of the endoplasmic reticulum Ca^2+^ store [22]. This gene negatively regulates the ER-stress response and positively regulates the stability of ATP6V1A and ATP1B1 subunits of V-ATPase by preventing their degradation through an unknown proteasome-independent mechanism [23]. Variants of the *WFS1* gene can cause a wide spectrum of disorders, ranging from an isolated hearing loss to a multiple-system disorder. The autosomal dominant isolated sensorineural NSHL (OMIM: #600965, DFNA6) occurs usually at low frequencies with moderate HL and is not a candidate for CI. However, cases with congenital severe or profound HL have also been described [16,21,22,24]. The autosomal recessive Wolfram syndrome type 1 (OMIM #222300, WFS) is a multisystem disorder, characterized by major symptoms (diabetes mellitus, diabetes insipidus, optic atrophy, seizures, and deafness) and additional clinical features (renal abnormalities, ataxia, dementia or intellectual disability, and diverse psychiatric illnesses) [25]. To *WFS1*-spectrum belongs also the autosomal dominant Wolfram-like syndrome, characterized by hearing loss and/or optic atrophy without the presence of all typical WFS-associated symptoms. Mutations associated with the recessive WFS are spread over the entire coding region of the *WFS1* gene and are typically inactivating or truncating, suggesting that the biallelic loss of protein function leads to a severe disease phenotype. In contrast, non-truncating/non-inactivating variants, most of which are located in the C-terminal protein domain, have been found in nonsyndromic HL families [26] or in Wolfram-like syndrome [25,27] *WFS1* variants, present in five patients in our study, were missense modifications, and three of these were located in the C-terminal domain. In one patient the diagnosis of an autosomal dominant Wolfram-like syndrome could be established, he carried a known pathogenic *WFS1* variant, Ala684Val, which has been described in several families with autosomal dominant congenital HL and a later onset optic atrophy [25]. Our patient was five years old at the age of genetic diagnosis and presented no ophthalmological symptoms yet. In the remaining four cases, an unambiguous causality of the *WFS1* variants could not be set up, based on the atypical phenotypes of the patients, the absence of a second variant *in trans*, or the missing segregation data. All but one of our patients with *WFS1* variants presented with severe to profound HL and three of them have already benefited from CI before the genetic testing. In the case of one patient, the brother with the same phenotype carried also the same *WFS1*-variant. Although only one heterozygous alteration was found in these five cases, WFS was still considered in the differential diagnosis, since deep intronic variants could not be detected with the methods used. These patients, especially those at a young age, should be carefully followed in case other symptoms associated with WFS or Wolfram-like syndrome develop later in life. 

The *MYO7A* gene encodes a protein classified as an unconventional myosin; which, is a motor molecule with structurally conserved heads that move along the actin filaments. Their highly divergent tails are presumed to be tethered to different macromolecular structures that move relative to actin filaments, thus enabling them to transport cargo [28]. Variants in the *MYO7A* gene might be responsible for autosomal dominant (OMIM #601317, DFNA11) and autosomal recessive deafness (OMIM #600060, DFNB2), as well as Usher syndrome (OMIM #276900, USH1), characterized by HL and retinopathy. In the autosomal dominant disease, the audiological phenotype can also be variable [2]. Missense variants in the motor domain of MYO7A protein usually cause a late onset of hearing loss with ascending tendency; while, variants in the tail domains are associated with a severe audiological phenotype. Intrafamilial heterogeneity has also been reported, [29]. Two variants in our patients were located in the motor domain and one variant in the tail domain (FERM2) of the protein. Furthermore, these patients presented no ophthalmologic symptoms, such as difficulty seeing at night or loss of vision, at the time of the genetic diagnosis (at the age of 2, 4, and 34 years). Considering the absence of a second variant *in trans*, the diagnosis of Usher syndrome or autosomal recessive HL is unlikely. However, a direct causality of the *MYO7A* variants could not be ultimately concluded. A close follow-up of these patients was recommended in case a retinopathy related to Usher syndrome should manifest later in life. 

X-linked inheritance accounts for less than 1–5% of NSHL cases [2]. We found one male patient (1%) with a possible disease-causing variant in the *POU3F4* gene. This gene encodes a member of the POU-III class of neural transcription factors, a family of proteins that play a role in inner ear development. The protein is thought to be involved in the mediation of epigenetic signals that induce striatal neuron–precursor differentiation [30]. Mutations in the *POU3F4* gene cause X-linked deafness-2 (OMIM #304400, DFNX2), also known as conductive deafness with stapes fixation. It is characterized by progressive conductive and sensorineural HL and a pathognomonic temporal bone deformity that includes dilation of the inner auditory canal and a fistulous connection between the internal auditory canal and the cochlear basal turn, resulting in a perilymphatic fluid “gusher” during stapes surgery [31,32,33].

Our diagnostic yield was 66% (92/139). A large study conducted by Sloan-Heggen and colleagues in 2016 [15], investigated 1,119 patients with NSHL and established an average 39% diagnostic yield (ranging from 22% to 50%, depending on the age of onset, familial occurrence, laterality, or inheritance pattern). On one hand, our detection rate was markedly higher; which, may be due to the selection of patients with bilateral NSHL and/or the abundance of severe or profound and prelingual NSHL in our cohort. However, our improved rate could also be due to the use of a more extensive gene panel. On the other hand, we have also detected inconclusive findings in 15% of the cases due to a missing second variant *in trans* or a genotype–phenotype discrepancy; which, ultimately points out the necessity of further genetic investigations for the unexplored DNA regions. The work by Shearer and colleagues [34] showed that as many as one in five patients with NSHL might carry causative CNVs. Thus, the detection rate of our experiment design could possibly be further improved by including genome-wide CNV analysis.

Genetic defects leading to HL in children may overlap with syndromic forms, which may account for up to 30% of prelingual deafness [1]. In addition, syndromic forms may mimic NSHL [15]. Given this complexity, when auditory symptoms are present before non-auditory symptoms develop, early genetic diagnosis of prelingual and postlingual HL could significantly improve the management of the disease for young patients.

The present study was the first to investigate nonsyndromic HL in a Hungarian population using a comprehensive stepwise genetic approach that included bidirectional capillary sequencing, MLPA, and an NGS panel of 108 HL genes. Our results showed that the implementation of this stepwise analysis is an improvement of simpler approaches, as our diagnostic yield was higher than usual and proved to be cost-effective as well. To further improve the diagnostic yield, the implementation of whole-exome sequencing may be taken into consideration. 

## 4. Materials and Methods

### 4.1. Patients

Patients with NSHL were recruited at the Department of Oto-Rhino-Laryngology and Head-Neck Surgery, Albert Szent-Györgyi Medical School, University of Szeged; Department of Oto-Rhino-Laryngology and Head-Neck Surgery, University of Pécs; and Department of Otorhinolaryngology and Head-Neck Surgery, University of Debrecen, between 2019 and 2020. Genetic testing was performed at the Department of Medical Genetics, Albert Szent-Györgyi Medical School, University of Szeged. The patients and/or their legal representatives gave their informed consent to the clinical study and genetic diagnostics. The study was approved by the local Ethical Committee (17427-6/2019/EÜIG) and was carried out according to the ethical guidelines and regulations of the Helsinki Declaration.

Overall, 139 non-related NSHL probands (75 females and 64 males) without a previous genetic diagnosis were enrolled in the study. The enrolled patients had already received cochlear implants before the genetic testing or were presented at the Department of Oto-Rhino-Laryngology to be evaluated for potential cochlea implantation. The age of the participants ranged from infancy to 68 years (mean age: 12.3 years). The enrolled patients were clinically diagnosed with bilateral sensorineural NSHL. In 2.7% of the cases, a familial form of HL could be deduced.

### 4.2. Clinical Characteristics of the Patients

The audiological analysis consisted of subjective and objective tests. Subjective hearing tests, such as pure-tone audiometry and speech recognition tests were used with adults and cooperative children. Objective hearing tests were performed for all patients, and included tympanometry, distortion product otoacoustic emissions (DPOEA), auditory brainstem response (ABR) and auditory steady-state response (ASSR). These objective tests were performed with normal middle ear ventilation (normal type A tympanogram on both sides). Little or no external hair cell activity could be registered by DPOAE, confirming inner ear origin in case of sensorineural HL. No signs of neural involvement were registered with ABR examination at high intensity, which is consistent with retrocochlear lesion.

With ASSR, the objective hearing threshold of patients was estimated to range from mild to total hearing loss: mild (>26), moderate (>41), moderately severe (>56), severe (>71 dB), and profound HL (>90). With the exception of the two patients with mild and four patients with moderate HL, all patients presented severe or profound HL, and thus a relatively homogenous cohort could be established for the genetic testing. The audiological results in the majority of the patients indicated no potential benefit from the traditional air-conduction hearing aid and thus, they were considered for CI. However, ultimately not all patients received CI (Appendix A).

Retrolabyrinthine damage of the cochlear nerve was detected with preoperative cerebral magnetic resonance imaging (using an inner ear protocol) and a high-resolution (0.4 mm slices) computed tomography scan of the middle and inner ear.

Patients were grouped into two categories based on their audiological history: prelingual HL (HL occurred in childhood before or during speech development) and postlingual HL (after speech development) (Appendix A).

### 4.3. Genetic Analysis

Genomic DNA was extracted from peripheral blood samples using the QIAamp DNA Blood Mini Kit (QIAGEN, Germany), as described in the manufacturer’s instructions. All samples were subjected to the stepwise molecular genetic analysis shown in Figure 1. 

All samples were first tested for variants in the *GJB2* (NM_004004.6)*, GJB3* (NM_024009.3)*,* and *GJB6* (NM_001110219.3) genes. The non-coding (exon 1) region of the *GJB2* gene was amplified as described previously [10]. The coding exons (exon 2) of all three *GJB* genes were amplified as two fragments and analyzed by bidirectional sequencing. Sequencing results were compared to the GRCh37/hg19 human reference genome. 

When no causative variant or only one heterozygous recessive *GJB2* variant was identified, the analysis was extended to the common *GJB6* deletions and multiplex ligation-dependent probe amplification (MLPA). Common *GJB6* deletions [del(GJB6-D13S1830) and del(GJB6-D13S1854)] were analyzed using the multiplex PCR assay designed by del Castillo and colleagues [14]. 

Del(GJB6-D13S1830) and del(GJB6-D13S1854) positive controls were kindly provided by Dr. Ignacio del Castillo, Unidad de Genética Molecular, Hospital Ramón y Cajal, Madrid, Spain. 

To detect further deletions and duplications, we used, according to the manufacturer’s instructions, the SALSA MLPA kit P163-C1 Hearing Loss probe mix (MRC-Holland, the Netherlands), which contains probes for all *GJB2* and *GJB6* exons and for *GJB3*, *POU3F4* and *WFS1* genes, and the SALSA MLPA Probemix P461 STRC-CATSPER2-OTOA (MRC-Holland, the Netherlands), which contains probes for deletions or duplications in the *STRC, CATSPER2* and *OTOA* genes as well as for gene conversions between *STRC* and its pseudogene *STRCP1*. Amplicon fragment length analysis was performed on an ABI 3500 Genetic Analyzer (Thermo Fisher Scientific, Waltham, MA, USA) and analyzed by Coffalyser.net software (v.220513.1739, MRC-Holland, Amsterdam, The Netherlands) (Appendix A). 

Patients with negative sequencing results for the *GJB2, GJB3* and *GJB6* genes, negative MLPA results or only one heterozygous *GJB2* allele were further analyzed with an NGS-based gene panel (Figure 1). A SureSelect custom target enrichment library was designed for 108 genes associated with NSHL using SureDesign software (Agilent Technologies, Santa Clara, CA, USA) (Appendix A). Library preparation was carried out using the SureSelectQXT Reagent Kit (Agilent Technologies, Santa Clara, CA, USA). Pooled libraries were sequenced on an Illumina NextSeq 550 NGS platform using the 300-cycles Mid Output Kit v2.5 (Illumina, Inc., San Diego, CA, USA). Adapter-trimmed and Q30-filtered paired-end reads were aligned to the hg19 Human Reference Genome using the Burrows–Wheeler Aligner (BWA). Duplicates were marked using the Picard software package (version: 1.79). The Genome Analysis Toolkit (GATK) was used for variant calling (BaseSpace BWA Enrichment Workflow v2.1.1. with BWA 0.7.7-isis-1.0.0, Picard: 1.79 and GATK v1.6-23-gf0210b3). On average, 99.18% of the target genes were covered more than twentyfold. Variants passed by the GATK filter were used for downstream analysis and annotated using the ANNOVAR software tool (v.2.0.2 (27 March 2019), [35]). Assuming an autosomal recessive inheritance, only variants were reported, whose allele frequency was below 1% at the time of analysis. Variants with an allele frequency below 0.1% were considered for a dominant inheritance pattern. The allele frequency in the general population was estimated using the Genome Aggregation Database v2.1.1 and v3.1.2 (gnomAD). The evaluation of identified sequence variants was based on public databases (HGMD [36], Decipher [37], ClinVar [38]). For variant filtering and interpretation, VarSome [39] and Franklin bioinformatic platforms (https://franklin.genoox.com, accessed on 22 January 2023) were used, incorporating the guidelines of the American College of Medical Genetics and Genomics (ACMG) [6]. For variant filtering and interpretation, VarSome [39] and Franklin bioinformatic platforms (https://franklin.genoox.com) were used, incorporating the guidelines of the American College of Medical Genetics and Genomics (ACMG). 

### 4.4. Statistical Analysis

GraphPad Prism, version 6.01 for Windows software (GraphPad Software, San Diego, CA, USA), was used for statistical analysis. The Chi-squared test with Yates’ correction was performed to compare the clinical characteristics between the prelingual and postlingual groups. *p* < 0.05 was considered to be statistically significant. 

## Figures and Tables

**Figure 1 ijms-24-07401-f001:**
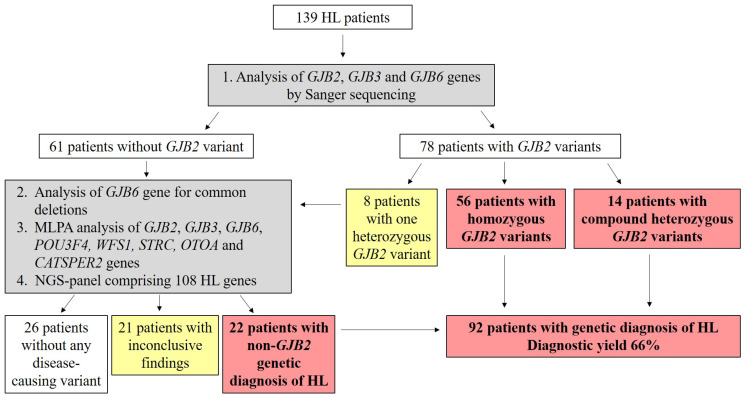
Workflow and major screening findings of the stepwise genetic analysis used with the Hungarian NSHL patient cohort. NSHL: nonsyndromic hearing loss; MLPA: multiplex ligation-dependent probe amplification; NGS: next-generation sequencing. Inconclusive findings include cases with only one heterozygous variant in autosomal recessive genes and cases with variants with both autosomal dominant and recessive inheritance; where, the genotype–phenotype comparison could not make a clear diagnosis.

**Figure 2 ijms-24-07401-f002:**
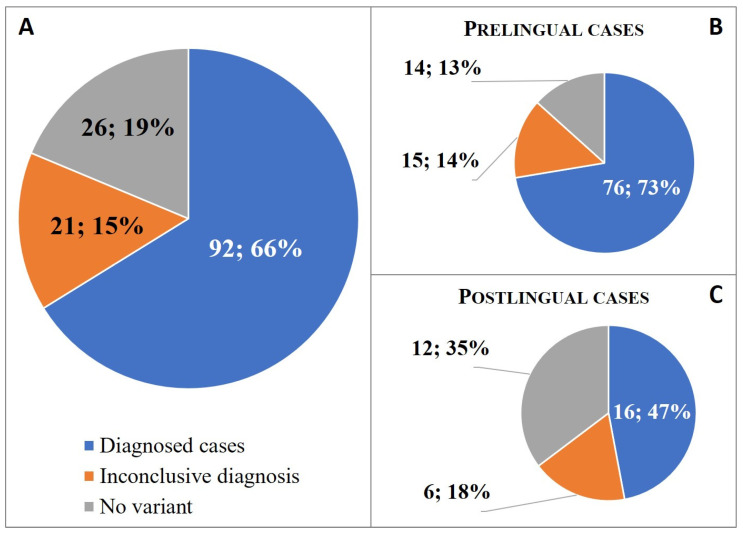
Distribution of diagnosis for patients in the Hungarian NSHL patient cohort. Numbers within the sections of the circular chart: number of patients; percentage of patients. (**A**) Complete (N = 139); (**B**) prelingual NSHL (N = 105); and (**C**) postlingual NSHL (N = 34) cohorts.

**Figure 3 ijms-24-07401-f003:**
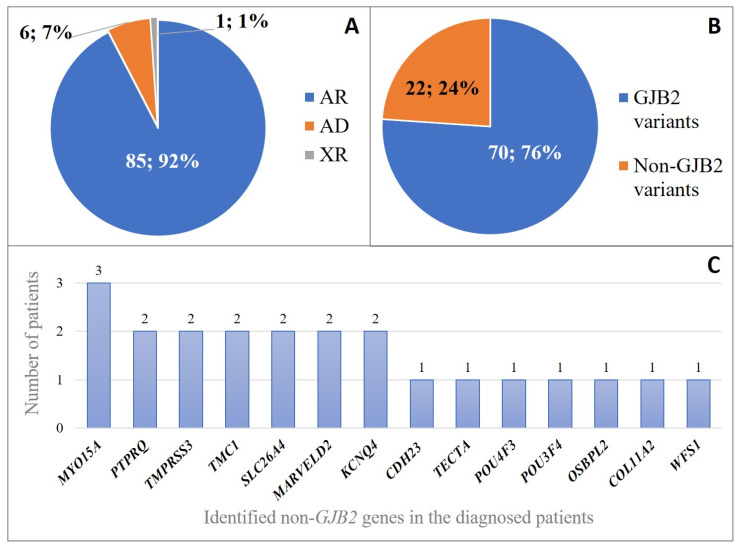
Distribution of inheritance patterns (**A**), *GJB2* (**B**), and non-*GJB2* variants (**C**) in the Hungarian NSHL patient cohort. Diagnosed patients: N = 92. Numbers within a section of a circular graph indicate the number of patients (percentage of patients). AR: autosomal recessive; AD: autosomal dominant; XR: X-linked recessive.

**Table 1 ijms-24-07401-t001:** Pathogenic *GJB2* variants identified in the Hungarian NSHL patient cohort.

Variant—HGVS	MAF	ACMG	Number of Cases (Homozygous/Compound Heterozygous/Heterozygous)
c.35delG; p.Gly12ValfsTer2	0.00597	Pathogenic ^a^	56/11/4
c.71G>A; p.Trp24Ter	0.00058	Pathogenic ^a^	0/0//1
c.101T>C; p.Met34Thr	0.00868	Pathogenic ^a^	0/1//2
c.109G>T; p.Val37Phe	0	Pathogenic ^a^	0/2/0
c.119C>A; p.Ala40Glu	0	Pathogenic ^b^	0/0//1
c.139G>T; p.Glu47Ter	0.00013	Pathogenic ^a^	0/1/0
c.167delT; p.Leu56ArgfsTer26	0.00089	Pathogenic ^a^	0/3/0
c.269T>C; p.Leu90Pro	0.00065	Pathogenic ^a^	0/1/0
c.313_326del; p.Lys105GlyfsTer5	0.00013	Pathogenic ^a^	0/2/0
c.427C>T; p.Arg143Trp	0.00012	Pathogenic	0/2/0
c.439G>A; p.Glu147Lys	0.00001	Pathogenic ^b^	0/1/0
c.551G>C; p.Arg184Pro	0.00006	Pathogenic	0/1/0
c.−23+1G>A (IVS1+1G>A)	0.00019	Pathogenic ^a^	0/3/0

HGVS: nomenclature from the Human Genome Variation Society; MAF: minor allele frequency from the gnomAD Database [5]; ACMG: variant classification according to the guidelines of the American College of Medical Genetics and Genomics [6]; ^a^ variant previously reported in a Hungarian cohort by Tóth et al. [7]; ^b^ variant previously reported in a Hungarian cohort by Kecskeméti et al. [8].

**Table 2 ijms-24-07401-t002:** Characteristics of rare variants identified in the Hungarian NSHL patient cohort.

Gene	Transcript (hg19)	Variant—HGVS (cDNA; Protein)	MAF	ACMG Classification	Inherit.	No of Alleles	No of Cases (hmz/Compound htz/htz)
** *CDH23* **	NM_022124	**c.6204del; p.Phe2069LeufsTer11**	0	LP	AR	1	0/0/1
** *CDH23* **	NM_022124	**c.1349_1350del; p.Leu450HisfsTer3**	0	LP	AR	1	0/1/0
** *CDH23* **	NM_022124	**c.4846-2A>G; p.?**	0	P	AR	1	0/1/0
** *COL11A2* **	NM_080680.3	**c.3403G>C; p.Gly1135Arg**	0	VUS/LP	AR/AD	1	0/1/0
** *COL11A2* **	NM_080680.3	**c.1297C>T; p.Arg433Ter**	0	LP	AR/AD	1	0/1/0
** *KCNQ4* **	NM_004700	**c.647G>C; p.Arg216Pro**	0	VUS	AD	1	0/0/1
** *KCNQ4* **	NM_004700	**c.1031_1040del; p.Asn344ArgfsTer11**	0	P	AD	1	0/0/1
** *MARVELD2* **	NM_001038603	c.1331+2T>C; p.?	0.0000398	P	AR	4	2/0/0
** *MYO15A* **	NM_016239	**c.2493_2505del; p.Arg832ProfsTer27**	0	LP	AR	1	0/1/0
** *MYO15A* **	NM_016239	c.2677C>T; p.Arg893Ter	0.0000211	P	AR	1	0/1/0
** *MYO15A* **	NM_016239	c.4030C>T; p.Gln1344Ter	0	P	AR	1	0/1/0
** *MYO15A* **	NM_016239	c.8183G>A; p.Arg2728His	0.000189	P	AR	1	0/1/0
** *MYO15A* **	NM_016239	**c.8153T>C; p.Leu2718Pro**	0	VUS/LP	AR	1	0/1/0
** *MYO15A* **	NM_016239	c.8548C>T; p.Arg2850Ter	0.00002	P	AR	1	0/1/0
** *MYO7A* **	NM_000260	c.268C>T; p.Arg90Trp	0.0000438	VUS/LP	AD/AR	1	0/0/1
** *MYO7A* **	NM_000260	**c.770G>T; p.Cys257Phe**	0	VUS	AD/AR	1	0/0/1
** *MYO7A* **	NM_000260	**c.4087G>A; p.Ala1363Thr**	0	VUS	AD/AR	1	0/0/1
** *OSBPL2* **	NM_001363878	**c.313C>T; p.His105Tyr**	0.0000159	VUS	AD	1	0/0/1
** *OTOF* **	NM_194248.3	**c.2665del; p.Leu889SerfsTer111**	0	LP	AR	1	0/0/1
** *POU3F4* **	NM_000307	**c.446G>T; p.Gly149Val**	0	VUS	XR	1	0/0/1
** *POU4F3* **	NM_002700	**c.868G>C; p.Glu290Gln**	0.00000398	VUS/LP	AD	1	0/0/1
** *PNPT1* **	NM_033109.5	**c.79delG; p.Asp27IlefsTer25**	0	LP	AR	1	0/0/1
** *PTPRQ* **	NM_001145026	**c.5959C>T; p.Gln1987Ter**	0	LP	AR	5	2/0/1
** *SLC26A4* **	NM_000441	c.349C>T; p.Leu117Phe	0.000326	LP	AR	1	0/1/0
** *SLC26A4* **	NM_000441	c.1204G>T; p.Val402Leu	0	P	AR	1	0/1/0
** *SLC26A4* **	NM_000441	c.1670G>T; p.Gly557Val	0.00000399	LP	AR	2	1/0/0
** *TECTA* **	NM_005422	**c.6094G>T; p.Asp2032Tyr**	0.00000398	VUS/LP	AD/AR	1	0/0/1
** *TJP2* **	NM_001170416	**c.53T>A; p.Leu18Ter**	0	VUS/LP	AD	1	0/0/1
** *TMC1* **	NM_138691	**c.312_325del; p.Val106MetfsTer8**	0	LP	AD/AR	1	0/1/0
** *TMC1* **	NM_138691	c.2030T>C; p.Ile677Tyr	0.0000119	VUS/LP	AD/AR	2	1/0/0
** *TMC1* **	NM_138691	c.2050G>A; p.Asp684Asn	0.0000239	VUS/LP	AD/AR	1	0/1/0
** *TMPRSS3* **	NM_001256317	c.208delC; p.His70ThrfsTer19	0.000489	P	AR	3	1/1/0
** *TMPRSS3* **	NM_001256317	c.646C>T; p.Arg216Cys	0.00002387	P	AR	1	0/1/0
** *USH1C* **	NM_153676.4	c.241C>T; p.Arg81Cys	0.00002001	VUS	AR	1	0/0/1
** *WFS1* **	NM_006005	c.958C>T; p.Pro320Ser	0	VUS	AD/AR	1	0/0/1
** *WFS1* **	NM_006005	c.1181A>T; p.Glu394Val	0.0001273	LP	AD/AR	1	0/0/1
** *WFS1* **	NM_006005	c.2051C>T; p.Ala684Val	0	P	AD/AR	1	0/0/1
** *WFS1* **	NM_006005	**c.2527A>G; p.Lys843Glu**	0	VUS/LP	AD/AR	1	0/0/1
** *WFS1* **	NM_006005	c.2575C>T; p.Arg859Trp	0.0000319	VUS	AD/AR	1	0/0/1
** *STRC* **	NM_153700	Contiguous gene deletion, including the *CKMT1B, STRC* and *CATSPER2* genes		P	AR	1	0/0/1
** *OTOA* **	NM_170664	Contiguous gene duplication, including the *METTL9* and *OTOA* genes		VUS	AR	1	0/0/1

HGVS: nomenclature from the Human Genome Variation Society; MAF: minor allele frequency from the gnomAD Database; ACMG: variant classification according to the guidelines of the American College of Medical Genetics and Genomics [6]; hmz: homozygous; htz: heterozygous; P: pathogenic; LP: likely pathogenic; VUS: variant of unknown significance; VUS/LP: variant of unknown significance with a pathogen tendency; AR: autosomal recessive; AD: autosomal dominant; XR: X-linked recessive. p.?: intronic variant with a splicing alteration. Novel variants are indicated in bold.

## Data Availability

Not applicable.

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
