# Peer review of "Genetic Etiology of Nonsyndromic Hearing Loss in Hungarian Patients"

_ijms, 2023, doi:10.3390/ijms24087401_

Round 1

Reviewer 1 Report

Pál et al. studied the genetic etiology of non-syndromic Hearing loss in 139 patients. The study is concise and well-written.

A few minor changes are required in the manuscript.

·      Did the author check the Allelic Frequency of the variants in all these databases, gnomAD (2.1 and 3.1 version), Bravo, all of US, and GME?

·      The other parameters of the variants (CADD score, GERP++) also need to add to the table (can be added in the supplementary table or in the manuscript main table).

·      Did the author verify the novel variant by segregation in the family?

·      How was the vestibular dysfunction excluded in the patients?

·      The variants were classified based on ACMG guidelines. Please mention in the table which criteria were fulfilled for the classification of the variants from ACMG guidelines.

(for example; PPS1-1 PP3, or PM2)

Author Response

Response to Reviewer 1

We thank Reviewer 1 for the careful reading and valuable comments, we revised the manuscript accordingly.

Question 1: Did the author check the Allelic Frequency of the variants in all these databases, gnomAD (2.1 and 3.1 version), Bravo, all of US, and GME?

Response 1:

We originally checked the allele frequency of the variants only in gnomAD 2.1. but added the data from Bravo, all of US, and GME in the Supplementary Table, as requested.

Question 2: The other parameters of the variants (CADD score, GERP++) also need to add to the table (can be added in the supplementary table or in the manuscript main table).

Response 2:

CADD scores for missense variants and GERP++ data, together with PhyloP100way have been added to the revised supplementary table.

Question 3: Did the author verify the novel variant by segregation in the family?

Response 3:

The segregation analysis of the variants was unfortunately not possible in all cases, this is why, some of the novel variants could not be better classified than VUS. In 2.7% of the cases, we could establish the diagnosis of familial HL, however, it is most probably underestimated.

Question 4: How was the vestibular dysfunction excluded in the patients?

Response 4:

Vestibular dysfunction was excluded by video head impulse test and Hallpike Caloric Test. Furthermore, MRI for the inner ear and high resolution CT was performed, especially in patients before CI. Thus, structural anomalies, such as enlarged vestibular aqueduct, could be also excluded.

Question 5: The variants were classified based on ACMG guidelines. Please mention in the table which criteria were fulfilled for the classification of the variants from ACMG guidelines. (for example; PPS1-1 PP3, or PM2)

Response 5:

The variant classification has been added to Supplementary Table.

Reviewer 2 Report

This study used a comprehensive, stepwise, sequencing-based genetic testing approach to screen Hungarian patients with NSHL and to develop the most effective screening scheme for this disease. The implementation of this stepwise alalysis markedly increased their diagnostic yield and proved to be cost-effective as well. This is logical description. I agreed to be accepted as it is.

Author Response

We thank Reviewer for the careful reading, the appreciative comments and the supporting evaluation.

Reviewer 3 Report

Well designed, conducted, and presented.

Accep as it is!

Author Response

(The authors gave the same response as above.)

Reviewer 4 Report

Comments and suggestions are in attachment. 

Round 2

Reviewer 4 Report

The authors revised the paper according to major suggestions, nevertheless, the data should be analyzed more carefully and manuscript should be rewritten.

Results should be reanalyzed in the context of clinical data and available literature and databases. The Authors added some clinical data but still do not properly interpret genotype-phenotype correlation. In the paragraph 8 of the Results section there is an information on Wolfram syndrome, but in the Discussion section the Authors clearly indicate the presence of the ADHL in patients with heterozygous variants in WFS1 gene.  This is a serious mistake. Reporting variants of unknown clinical significance, particularly as a cause of ADHL, requires caution. In such cases, it is worth to include pedigrees so that the reader can critically evaluate the results. Variants in genes that, in addition to ADHL, also cause ARHL or syndromic forms of HL should be verified in the families of patients - this is undeniable evidence of their probable pathogenicity. In the reviewed manuscript, the Authors report the variants as possibly causative even if they are associated with a different type of inheritance - including c.1181A>T variant in the WFS1 gene, which is most likely associated with Wolfram syndrome (ClinVar ID RCV001644744.1) and not ADHL. In the gnomAD database, it occurs 34 times in the heterozygous state - too often for a dominantly inherited disease. The other WFS1 variant – c.2051C>T – is well known mutation involved in the development of the Wolfram-like syndrome (PMID: 21538838, 22238590) and not in ADHL. The Authors have completely forgotten about this syndromic form of HL caused by WFS1 variants. More severe HL observed in children is more frequently identified in Wolfram-like syndrome than in patients with WFS1-related ADHL. Similar concerns can also be raised with variants located in other genes, including MYO7A.

Other comments:

Information provided in lines 101-103 are duplicated in lines 135-137. Similar duplication is observed in lines 447-452.

Please check again which variants are novel and which are known – e.g. SLC26A4 c.1670G>T has been described in PMID: 36833263

Author Response

The authors revised the paper according to major suggestions, nevertheless, the data should be analyzed more carefully and manuscript should be rewritten.

Results should be reanalyzed in the context of clinical data and available literature and databases. The Authors added some clinical data but still do not properly interpret genotype-phenotype correlation. In the paragraph 8 of the Results section there is an information on Wolfram syndrome, but in the Discussion section the Authors clearly indicate the presence of the ADHL in patients with heterozygous variants in WFS1 gene.  This is a serious mistake. Reporting variants of unknown clinical significance, particularly as a cause of ADHL, requires caution. In such cases, it is worth to include pedigrees so that the reader can critically evaluate the results. Variants in genes that, in addition to ADHL, also cause ARHL or syndromic forms of HL should be verified in the families of patients - this is undeniable evidence of their probable pathogenicity. In the reviewed manuscript, the Authors report the variants as possibly causative even if they are associated with a different type of inheritance - including c.1181A>T variant in the WFS1 gene, which is most likely associated with Wolfram syndrome (ClinVar ID RCV001644744.1) and not ADHL. In the gnomAD database, it occurs 34 times in the heterozygous state - too often for a dominantly inherited disease. The other WFS1 variant – c.2051C>T – is well known mutation involved in the development of the Wolfram-like syndrome (PMID: 21538838, 22238590) and not in ADHL. The Authors have completely forgotten about this syndromic form of HL caused by WFS1 variants. More severe HL observed in children is more frequently identified in Wolfram-like syndrome than in patients with WFS1-related ADHL. Similar concerns can also be raised with variants located in other genes, including MYO7A.

We thank reviewer 4 for the thorough comments. We have rewritten the manuscript. Please see section Results and Discussions.

Other comments:

Information provided in lines 101-103 are duplicated in lines 135-137. Similar duplication is observed in lines 447-452.

We thank reviewer 4 the comment. We have deleted the duplication.

Please check again which variants are novel and which are known – e.g. SLC26A4 c.1670G>T has been described in PMID: 36833263

We thank reviewer 4 the comment. We were not aware of this publication, since the variant interpretation and classification was carried out at the beginning of January 2023 and the above-mentioned article was published on 28th January 2023. We have revised the manuscript and datasets accordingly. Furthermore, we have made an update on all variants, as requested, by using Mastermind Search on 29th March 2023. We found no further discrepancy.

Round 3

Reviewer 4 Report

Thank you for all the corrections. The article has been sufficiently improved.